# Diffusion Compose: Compositional Depth Aware Scene Editing in Diffusion Models

## Abstract

We introduce Diffusion Compose, a zero-shot approach for depth-aware scene editing using Text-to-Image diffusion models. While existing methods for 3D-aware editing focus on object-centric control, they do not support compositional depth-aware edits, such as placing objects at specific depths or combining multiple scenes realistically. We address this by incorporating depth-based multiplane scene representation in diffusion models. These planes, placed at fixed depths, can be individually edited or composed to enable 3D-aware scene modifications. However, direct manipulation of multiplane representation of diffusion latents often leads to identity loss or unrealistic blending. To overcome this, we propose a novel multiplane feature guidance technique that gradually aligns source latents with the target edit at each denoising step. We validate Diffusion Compose on two challenging tasks: a) scene composition, blending scenes with consistent depth order and scene illumination, and b) depth-aware object insertion, inserting novel objects at specified depths in a scene while preserving occlusions and scene structure and illumination. Extensive experiments demonstrate that Diffusion Compose significantly outperforms task-specific baselines for object placement and harmonization. A user study further confirms that it produces realistic, identity-preserving, and accurate depth-aware scene edits.

## 1 Introduction

Text-to-Image (T2I) diffusion models Rombach et al. (2022); Saharia et al. (2022); Esser et al. (2024) can generate highly realistic images from text prompts. Various conditioning mechanisms have been proposed Zhang et al. (2023a); Epstein et al. (2023) for complex image editing, such as altering scene appearance Brooks et al. (2023) or teleporting objects within scenes Chen et al. (2024). However, these approaches lack the ability to edit scenes with 3D control, such as placing a *new* vase at a specific 3D location in an image (Fig. 1). Achieving this requires addressing the following challenges: **i) Geometric consistency:** the placed object should fit naturally in the scene **ii) Occlusion handling:** for realistic placement, the placed object should be naturally occluded by the existing objects without any artifacts **iii) Illumination and lighting:** the placed object should respect the lighting in the scene to create realistic shading.

Existing methods for 3D control in T2I models primarily focus on editing geometric object properties such as rotating or translating the existing objects in the scene. This is achieved by applying the required geometric transformation to the diffusion features of the individual objects during denoising Wang et al. (2024); Pandey et al. (2024a); Sajnani et al. (2024); Kumari et al. (2024). Others rely on large-scale training with synthetic datasets conditioning on explicit 3D pose or geometric information with text Michel et al. (2024a); Wu et al. (2024), but struggle with generalization to real-world scenes. Although effective for object-centric 3D editing, these methods lack the ability to perform *compositional* 3D scene editing, such as depth-aware object insertion or scene composition. Our framework addresses this gap, enabling designers and artists to achieve precise object placement and seamless scene blending with depth-aware control. This enables workflows in areas such as advertising, game design, and visual effects, where depth-based layering is crucial.

We propose *DiffusionCompose* to enable depth-aware scene editing from a single image without the need to explicitly model a complete scene geometry. The key idea is to use a multiplane representation, where planes are placed at discrete depth levels, allowing for 3D-aware editing by manipulating

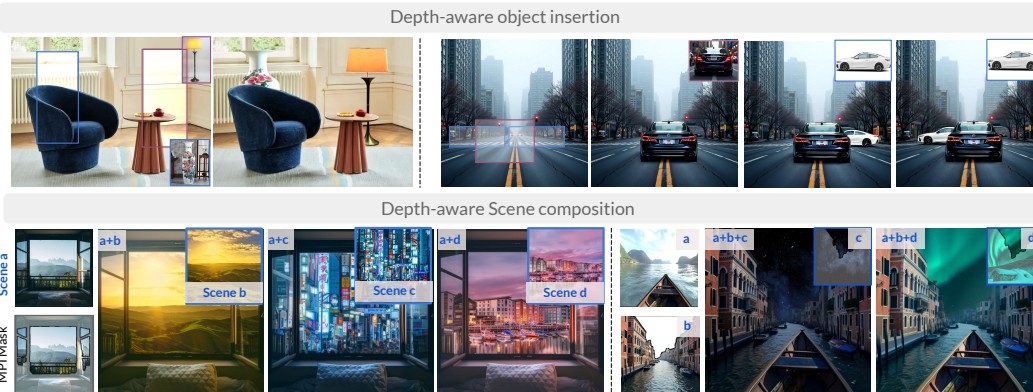

Figure 1: *Diffusion Compose* enables zero-shot depth-aware editing in real images: **i)** realistic object insertion in 3D handling complex scene effects, such as generating realistic occlusions for the vase while preserving object identity **ii)** depth-aware composition of multiple scenes with intro scene interactions such as illumination changes on the pillow from the foreground.

individual planes. We integrate this representation into diffusion models at inference time to achieve realistic *zero-shot* depth-aware edits. Directly applying multiplane representation to the latent space at intermediate timesteps leads to inferior results, suffering from preserving scene content or incorrect scene illumination. To address this, we introduce a novel *multiplane feature guidance*, which gradually guides the latents toward the target edit during each denoising step. Specifically, we align the multiplane representations of the intermediate diffusion U-Net features from the source and target edit latents, while preserving the distribution of the pretrained T2I model. This softer way to guide enables high-quality depth-aware edits with consistent geometry while respecting occlusions and scene illumination (Fig. 1).

Our *zero-shot* approach enables highly realistic scene edits by leveraging the rich priors of T2I models. We demonstrate the effectiveness of our framework via two challenging depth-aware editing tasks - **i)** *object insertion*, where novel objects are inserted at user-defined depths with proper occlusion and illumination, and **ii)** *scene composition:* depth-aware composition of multiple scenes with consistent scene illumination. Extensive experiments and a user study demonstrate the effectiveness of our method. For a comprehensive evaluation, we curated a test dataset of complex scenes and outperform existing object placement and scene harmonization baselines, without explicitly training for these tasks.

**Our major contributions** are threefold: **i)** first-of-its-kind zero-shot approach for depth-aware scene editing by integrating multiplane representations in Text-to-Image diffusion models. **ii)** novel *multiplane feature guidance* to slowly update the intermediate diffusion features for realistic depth-aware editing. **iii)** application of multiplane feature guidance to solve the challenging task of depth-aware object insertion and scene composition, with consistent occlusion and scene illumination.

## 2 RELATED WORKS

**Editing in Generative models.** Text-to-Image diffusion models have enabled several image editing tasks previously difficult to achieve Hertz et al. (2022); Epstein et al. (2023); Pandey et al. (2024a); Brooks et al. (2023). An effective approach for editing with diffusion models involves manipulating the intermediate cross-attention and self-attention maps Hertz et al. (2022); Patashnik et al. (2023); Cao et al. (2023) as they provide control in defining the layout, structure, and color in an image. This operation can be performed during inference time, eliminating the need for training. Some methods swap the attention maps Hertz et al. (2022); Cao et al. (2023), and while concatenate both features and take attention across the batch Zhou et al. (2024); Tewel et al. (2024). Control-Net Zhang et al. (2023a) introduced spatial conditioning modalities such as depth maps or edge maps for finer control. Another set of works explores the diffusion framework and condition at different timestepPatashnik et al. (2023); Zhang et al. (2023b) at U-Net layersVoynov et al. (2023b); Alaluf et al. (2023). Others aim to find semantic direction in latent space or the text space Kwon et al. (2022); Brack et al. (2023); Baumann et al. (2024) for editing. However, these approaches do not allow for 3D control in the generated scene.

**3D editing with Generative Models.** Diffusion models though excellent at generating realistic images often fail to generate consistent 3D effects Sarkar et al. (2024); Upadhyay (2024). To achieve some 3D control in the generation a common approach is to use depth conditioned diffusion model and edit the depth map. One effective approach is to provide guidance Mou et al. (2024); Pandey et al. (2024b) or lift the 2D diffusion feature to intermediate 3D representation Pandey et al. (2024a); Sajnani et al. (2024) using depth and edit it, and use diffusion models to refine the rendered image Wang et al. (2024); Yenphraphai et al. (2024); Michel et al. (2024b)or perform large scale finetuning with 3D conditioned dataset Bhat et al. (2024); Michel et al. (2024b). In Wang et al. (2024), multiple iterations of 3D edits are performed in image space followed by image refinement using diffusion prior. Similarly, in Pandey et al. (2024b), edited depth maps are used for conditioning and performing appearance guidance to preserve object and scene identity. On the other hand, Bhat et al. (2024); Michel et al. (2024a); Wu et al. (2024) perform large-scale training to achieve object-centric 3D geometric control, however struggle to handle complex real scenes with multiple objects.

**Object Insertion.** Given a 3D representation one can insert an object while following scene geometry Shahbazi et al. (2024) and perform 3D aware edits Haque et al. (2023). However, obtaining a good 3D representation of a scene from a single image is difficult. In Ge et al. (2024), they find an approximate floor plane and scene lighting to generate and place synthetic 3D assets in the given scene with relighting which is challenging to obtain for real-world objects. When dealing with only a single scene image and object image, the most common methods for object placement are reference-based inpainting methods like IP-Adapter, PaintbyExample, and Anydoor Ye et al. (2023); Yang et al. (2023); Chen et al. (2024). However, these methods do not provide control to place objects at a particular depth and always generate full objects without occlusion. Another recent work Winter et al. (2024) performs realistic object insertion with accurate lighting and shading as it is trained on high-quality datasets curated for the task. In this work, we propose a zero-shot approach to generate realistic depth-aware object placement given a single object and background image with consistent shadings and blending effects.

## 3 METHOD

### 3.1 OVERVIEW

Our goal is to perform realistic depth-aware editing with a single image using the generative priors of Text-to-Image models without retraining. To this end, we utilize the multiplane scene representation, where a scene is decomposed into a set of frontoparallel planes at fixed depths, enabling 3D scene editing. Directly applied in the image space the multiplane representation does not respect the scene semantics during editing and leads to 'cut-paste' appearance (Fig. 3). Instead, we integrate the multiplane representation into the latent space of T2I diffusion models, capitalizing on their rich image generation priors. We accomplish this through *multiplane feature guidance* at each denoising step, updating the intermediate source latents to enable consistent depth-aware editing. In the following sections, we discuss the preliminaries of our work and present our approach for multiplane feature guidance, along with its applications in depth-aware editing.

### 3.2 PRELIMINARIES

**Diffusion models** learn to transform random noise into an image with iterative denoising. In the forward diffusion process, image $x_0$ is corrupted by sequentially adding standard Gaussian noise $\epsilon$. A denoiser network $\epsilon_\theta$ is trained to estimate the added noise, conditioned on the timestep and optional conditioning such as text. For generating images, the reverse diffusion process denoises the random noise $x_T$, with several passes through denoising network $\epsilon_\theta$. To accelerate the diffusion models, Latent Diffusion Models Rombach et al. (2022) take a two-stage approach where the input image is first encoded into a lower dimensional latent space, and the diffusion process is applied in the compressed latent space, significantly reducing the computational requirements.

**Guidance.** There are two main approaches for conditioning diffusion models on additional modalities: *classifier guidance* and *classifier-free guidance*. In classifier-free guidance Ho & Salimans (2022), conditional $\epsilon_\theta(x_t, y, t)$ predictions with $y$ conditioning are combined with unconditional predictions $\epsilon_\theta(x_t, t)$ using a scalar weight $w(t)$. Classifier guidance, on the other hand, provides inference time conditioning by guiding the reverse diffusion process using a predefined energy func-

Image MPI | Latent MPI T = 10 | Latent MPI T = 20 | Latent MPI T = 30 | Latent MPI T = 40 | MP Feature Guidance

Figure 3: Ablation on depth-aware scene composition. a) Compositing a scene with multiplane in the image space (MPI) generates unnatural 'cut-paste' compositions, as it does not have semantic information, b) Using multiplane representation in the latent space of diffusion for scene composition, has a tradeoff between identity preservation of the scene contents ($\tau = 40$), and realism of the composition ($\tau = 10$) depending on the blending timestep. Our Multiplane feature guidance achieves realistic composition while preserving the structure from both scenes with interactions between scenes such as illumination changes on the bus from the background.

tion. For example, to generate class conditioned generation Dhariwal & Nichol (2021) defined the energy as the cross-entropy loss between the pretrained classifier's prediction $f(x_t)$ and the given class $y$. During generation, the predicted noise $\epsilon_\theta$ is adjusted to minimize the classifier loss $\mathcal{L}$, with $\lambda$ as the classifier guidance weight as follows:

$$\tilde{\epsilon}_\theta(x_t, t) = \epsilon_\theta(x_t, t) + \lambda \nabla_{x_t} \mathcal{L}(f(x_t), y) \qquad (1)$$

Several guidance approaches have been proposed to achieve inference time conditioning on sketch Voynov et al. (2023a), layout Bansal et al. (2023); Epstein et al. (2023), features Pandey et al. (2024a), opticalflow Geng & Owens and attribute distribution Parihar et al. (2024).

### 3.3 MULTIPLANE LATENT REPRESENTATION FOR TEXT-TO-IMAGE MODELS.

Multi-Plane Imaging (MPI) Shade et al. (1998); Szeliski & Golland (1999) is an effective $2.5D$ scene representation, where an image $x$ is factorized into $D$ frontoparallel planes in the camera frustum (Fig. 2). These planes are arranged at fixed depths $d_1 = d_{near}$ to $d_D = d_{far}$. Each plane is represented as an RGBA image with color $c_i$ and an opacity $\alpha_i$ for $i^{th}$ plane, each having a resolution $HxW$.

$$f(x) = \{(\alpha^1, c^1), (\alpha^2, c^2), .....(\alpha^D, c^D)\} \qquad (2)$$

Given an input image $x$, and corresponding depth map $x^{depth}$, we can construct the multiplane

representation $f$ by first discretizing the depth maps based on the predefined plane depths $d_1$ to $d_D$. Next, the discretized depth image can be decomposed into multiple opacity masks ($\alpha^i$), one for each discrete depth value. The color $c^i$ for each plane can be extracted by masking our region from $x$ using $\alpha^i$ i.e., $c^i = \alpha^i \cdot x$. After editing individual planes to $c^{i'}$, the image can be recomposed from the multiplane representation using the following::

$$\hat{x} = \sum_{i=1}^{D} (\alpha^i c^{i'} \cdot \prod_{j=i+1}^{D} (1 - \alpha^j)) \qquad (3)$$

Though the MPI representation is efficient, using it directly in the image space results in unnatural 3D edits as it does not handle complex scene effects such as geometric consistency and illumination.

**Scene Composition.** We implement the multiplane representation in the internals of diffusion models for generating realistic and depth-aware

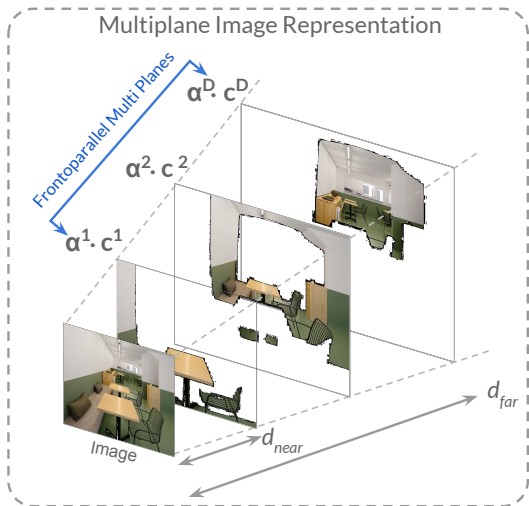

Figure 2: A given image can be represented as a set of RGBA planes placed at a fixed depth from $d_{near}$ to $d_{far}$, where $c^i$ is the RGB component and $\alpha_i$ is the opacity for layer $i$.

scene editing. We explain our approach with a running example of composing two scenes ($x^A$ and $x^B$) in a depth-aware manner. Precisely, we wish to realistically compose the foreground regions (depth $d_1$ to $d_k$) from scene $x^A$ and the background regions ($d_{k+1}$ to $d_D$) from scene $x^B$. One

approach is to first invert the two scenes using DDIM into their latent representation $z_{1:T}^A$ and $z_{1:T}^B$ and then combine the two latent representations for a timestep $\tau$ using corresponding multiplane representation. Specifically, we can generate multiplane representation of the latents $z_\tau^A$ and $z_\tau^B$ using downsampled depth maps of $x_A$ and $x_B$; yielding $f(z_\tau^A) = \{(\alpha^{A,1}, z_\tau^{A,1}), ..., (\alpha^{A,D}, z_\tau^{A,D})\}$ and $f(z_\tau^B) = \{(\alpha^{B,1}, z_\tau^{B,1}), ..., (\alpha^{B,D}, z_\tau^{B,D})\}$. For composing the two scenes at a user-provided depth value $d_{border}$, we can obtain the *largest* plane index $k$, having a lesser depth than $d_{border}$ (i.e. $d_k < d_{border}$). Next, we can fuse the two multiplane representations such that the first $k$ planes are from scene A and the remaining planes are from scene B to obtain a fused multiplane representation of edited latent $z_\tau^{edit}$, as:

$$f(z_\tau^{edit}) = \{(\alpha^{A,1}, z_\tau^{A,1}), ..., (\alpha^{A,k}, z_\tau^{A,k}), (\alpha^{B,k+1}, z_\tau^{B,k+1}), ..., (\alpha^{B,D}, z_\tau^{B,D})\} \qquad (4)$$

Finally, we can recompose the fused multiplane representation using Eq.3, to obtain the edited intermediate latent code $z_\tau^{edit}$. The edited latent $z_\tau^{edit}$ can then be denoised with diffusion model for the remaining $\tau$ timesteps, allowing for realistic blending of the scene Meng et al. (2021). Though this framework seems promising, it is ineffective in generating plausible scene composition as there is a tradeoff between realistic blending with complex scene effects and preserving scene content as shown in Fig. 3. A small $\tau$ does not provide enough freedom to recover the complex scene effects with denoising, and a large $\tau$ generates plausible composition but changes the scene contents significantly. To address this tradeoff, we propose a softer way of fusion via *multiplane feature guidance* Voynov et al. (2023a); Pandey et al. (2024a) that slowly nudge the generating latent to the target edit latent, while preserving the scene identity and generating complex scene effects.

**Multiplane Feature Guidance.** We start the generation process by sampling $z_T^{edit} \in \mathcal{N}(0, I)$ and then guiding intermediate latent $z_t^{edit}$ at each step of denoising. Given the intermediate scene latents $z_t^A$ and $z_t^B$ and the generating latent $z_t^{edit}$ at timestep $t$, we extract their corresponding diffusion U-Net features, $\Psi_{i,t}^A$, $\Psi_{i,t}^B$ and $\Psi_{i,t}^{edit}$, where $i$ is the diffusion model layer index. Next, we define a loss between the multiplane representation of the three features - $f(\Psi_{i,t}^A) = \{(\alpha^{A,j}, \psi_{i,t}^{A,j})\}_{j=1}^D$, $f(\Psi_{i,t}^B) = \{(\alpha^{B,j}, \psi_{i,t}^{B,j})\}_{j=1}^D$ and $f(\Psi_{i,t}^{edit}) = \{(\alpha^{edit,j}, \psi_{i,t}^{edit,j})\}_{j=1}^D$ for guidance.

*Intuition: We force the **initial planes** (1 to k) of $\Psi_{i,t}^{edit}$ to be close to initial planes of $\Psi_{i,t}^A$ and the **later planes** (k + 1 to D) of $\Psi_{i,t}^{edit}$ to be close to the later planes of $\Psi_{i,t}^B$ as shown in Fig. 4 To this end, we define the following guidance loss $\mathcal{L}(\Psi_t^A, \Psi_t^B, \Psi_t^{edit})$:*

$$\mathcal{L}(\Psi_t^A, \Psi_t^B, \Psi_t^{edit}) = \sum_i \left( \sum_{j=1}^k ||\psi_{i,t}^{A,j} - \psi_{i,t}^{edit,j}||^2 + \sum_{j=k+1}^D ||\psi_{i,t}^{B,j} - \psi_{i,t}^{edit,j}||^2 \right) \qquad (5)$$

We use the guidance loss to update $z_t^{edit}$ by computing $\nabla_{z_t^{edit}} \mathcal{L}(\Psi_t^A, \Psi_t^B, \Psi_t^{edit})$ for a few iterations at each denoising timestep. The feature guidance approach keeps the intermediate latents of the edited image in the training distribution of the pretrained model allowing high-quality generations. As shown in Fig. 1& 3, our method produces realistic scene compositions with complex lighting and effects, while preserving scene structure.

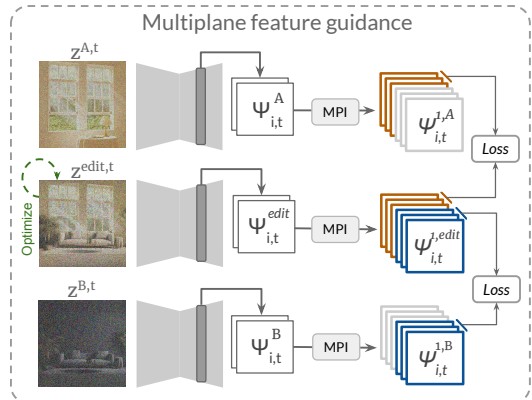

Figure 4: Multiplane Feature Guidance: At each denoising timestep the $t$ the multiplane diffusion features of the generated latent $z^{edit,t}$ are guided with the multiplane features of the input scene. The multiplane representation is obtained using the depth map of the inputs.

### 3.4 DEPTH AWARE OBJECT INSERTION

We show the application of *multiplane feature guidance* for solving the task of 3D-aware object placement in scenes. The inputs are the following: *a background image* $\mathbf{x}$, *reference object image* $\mathbf{x_o}$, *a 2D bounding box* $\mathbf{b}$, and *a depth value* $\mathbf{d_o}$ for placing the object. Instead of providing $\mathbf{d_o}$ explicitly, the relative depth of the object with respect to other objects can also be given (e.g. putting an object behind the table). The output is the background scene with plausible object placement, seamlessly blending with correct occlusions and consistent scene

illumination. For this task, we use state-of-the-art diffusion-based inpainting model $\mathcal{H}$ Chen et al. (2024) given its strong prior for object inpainting. However, standard inpainting models are designed to generate complete objects and struggle with depth-aware placement, particularly in scenes with significant occlusions (Fig. 5). We start by sampling $z_T^{edit} \in \mathcal{N}(0, I)$ and denoise it with $x$, $x_o$ and $b$ as additional conditioning in the reference-based conditioning model $\mathcal{H}$. During denoising, we update $z_t^{edit}$ with *multiplane feature guidance* for depth-aware object insertion. Specifically, given $d_o$, we obtain the first plane index $k$, such that $d_o < d_k$ and apply the guidance on deeper planes.

*Intuition:* *For depth-aware placement, information in all the **planes with lesser depth** ($i < k$) **should be preserved** from the background image and **planes with larger depth** ($i > k$) **are allowed to change**. To implement this, we apply multiple feature guidance on all the planes with index $i < k$ to update the current $z_t^{edit}$. Mathematically given features $\Psi^{edit}$ of generating image from $\mathcal{H}$ and $\Psi$ from DDIM inversion of background image $x$, we use the following loss function for guidance:*

$$\mathcal{L}(\Psi_t, \Psi_t^{edit}) = \sum_i \sum_{j=1}^{j=k} ||\psi_{i,t}^j - \psi_{i,t}^{edit,j}||^2 \tag{6}$$

Empirically, we find that applying Eq.4 at an intermediate timestep $\tau$, followed by guidance from Eq.6 at later timesteps, yields the most realistic object placement. This approach allows $\mathcal{H}$ to better interpret object depth in cases of occlusion. As a result, the generated object blends seamlessly in the scene respecting the occlusion and scene illuminations well while respecting occlusions and shadows (Fig. 5).

## 4 EXPERIMENTS

We perform extensive experiments to evaluate Diffusion Compose for the task of depth-aware editing. In this section, we first discuss the implementation and dataset details, followed by experiments on object placement scene composition, and ablations.

**Implementation Details.** We use Stable Diffusion v2-depth Rombach et al. (2022) which has depth conditioning as the base T2I model for scene composition and Anydoor Chen et al. (2024) for depth-aware object placement. For multiplane feature guidance, we use features from the last and the penultimate layers of the diffusion UNet which results in accurate edits as discussed in ablations. We give guidance from 0 to 38 timesteps for scene composition and update the latent $z_t^{edit}$ for 5 iterations at each timestep, and for object placement, we give guidance from 30 to 50 timestep and update the latent $z_t^{edit}$ for 3 iterations at each timestep. More details are in the appendix.

**Dataset.** As a zero-shot approach, we curated two datasets for a thorough evaluation of depth-aware editing tasks. For object placement, we compiled 490 image-object pairs annotated with 2D bounding boxes, object depth, and scene depth maps. The dataset includes diverse objects from indoor and outdoor environments, ensuring occlusion by other objects to effectively assess depth-aware placement. For scene composition, we curated a dataset of $2,844$ image pairs with diverse foreground and background scenes, combining the SSHarmonization dataset Jiang et al. (2021) and web-sourced images. The dataset spans a wide range of indoor and outdoor scenes, varying in lighting, composition, and appearance.

### 4.1 OBJECT PLACEMENT

To our knowledge, we are the first to perform depth-aware object placement using only a single object and background image. For the evaluation of Diffusion Compose, we define the following baselines for Diffusion Compose.

**Image MPI + Harmonization (Image MPI+H).** We perform MPI decomposition using the depth image of the background and paste the object in the next plane to the given object depth (Sec. 3.4). We preprocess the object image by segmenting the object of interest using SAM Kirillov et al. (2023) and resizing it according to the bounding box. Then, we recompose the MPI representation using Eq.3 to obtain the edited image. Additionally, to blend the object well in the scene, we apply a recent image harmonization technique Ke et al. (2022) on the placed object.

**Reference-based Inpainting.** We use state-of-the-art reference conditioned inpainting methods IP-Adapter Ye et al. (2023), Paint by example Yang et al. (2023), and Anydoor Chen et al. (2024) to

Figure 5: a) Results for depth-aware object placement. b) Our method can also place the given object at multiple locations in a depth-consistent manner. c) Comparison of depth-aware object placement: Image MPI + Harmonization results in an unnatural 'cut-paste' appearance for the inserted object. Inpainting models IP-adapter and Paint by example struggle to insert objects with consistent identity given the amodal bounding box. Anydoor achieves decent placement but has significant artifacts at the mask border (marked in red). Our method achieves realistic object placement while preserving the object identity and scene consistency.

inpaint the given bounding box with object image in the background scene. All these methods take a bounding box as input and place the object without considering occlusions. For a fair comparison, we adapt them for depth-aware placement, by using the MPI representation and masking the bounding box with the occluding objects (Fig. 5) to obtain an amodal bounding box. This will preserve the foreground regions during inpainting and give us the illusion that the object is placed behind other objects.

**Metrics.** We evaluate the object identity, realism of the output, and correctness of the object placement location in 3D. We use **DINO** Caron et al. (2021) feature similarity between the generated object in the bounding box and the reference object to measure identity preservation. To measure image realism, we compute **KID** Bińkowski et al. (2018) against COCO dataset Lin et al. (2014) as our evaluation set is relatively smaller to compute FID. To evaluate whether the object is actually placed, we use

Table 1: Comparison for depth-aware object placement. KID and $\Delta$ **depth** are shown in x$10^2$ units

| Method | DINO-sim ↑ | KID ↓ | $\Delta$ depth ↓ | Clip-sim ↑ |
|---|---|---|---|---|
| Image MPI + H | 0.576 | 4.7 | 2.985 | 68.5 |
| IP-Adapter | 0.244 | 5.3 | 9.366 | 27.81 |
| Paint by example | 0.273 | 4.9 | 6.733 | 60.12 |
| Anydoor | 0.507 | 4.9 | 3.176 | 83.23 |
| Ours | 0.545 | 4.8 | 2.989 | 84.86 |

CLIP Radford et al. (2021) similarity (**CLIP**) between *'a photo of object-name'* and the cropped bounding box from the generated image. If the object is generated correctly, the CLIP score should be higher. To assess depth consistency, we compute the discrepancy between the predicted object depth and the input placement depth. We estimate the depth of the generated image, compute the mean depth of the placed object and report normalized $\Delta$ **depth** across the dataset, with smaller values indicating more consistent depth-aware placement.

**Analysis.** We present the results for depth-aware object placement in Fig. 5, and Tab. 1. Image MPI + Harmonization generates consistent scene lighting for objects and identity-preserving placement but results in a *copy-paste* appearance and generates physically implausible compositions, such as a tilted vase and teddy hanging in the air also quantified with our user study Sec. 4.3. The reference conditioned inpainting models IP-adapter and Paint by example, struggle to generate accurate ob-

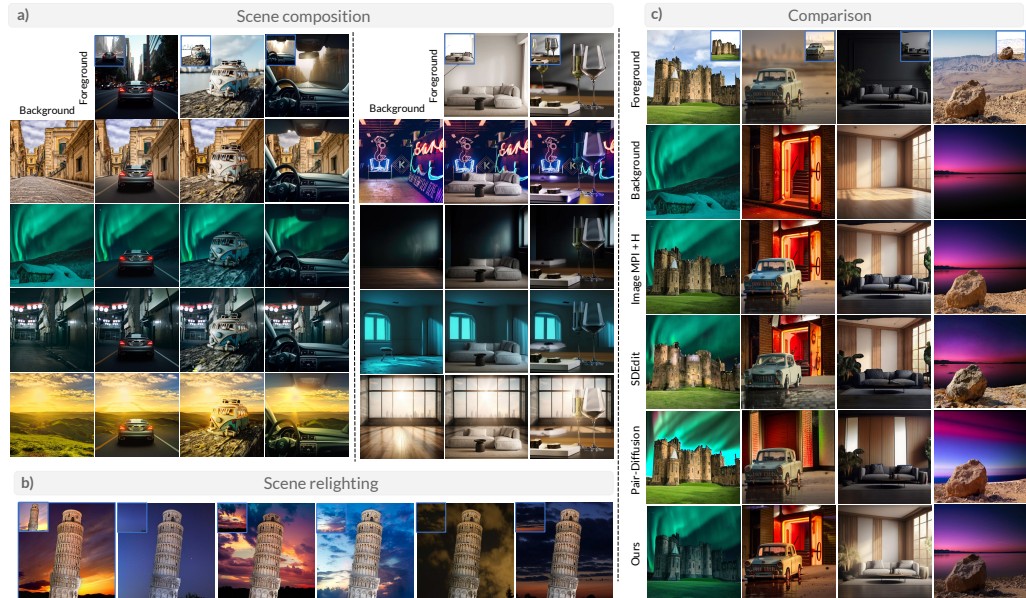

Figure 6: a) Results for scene composition b) Given a foreground scene, we can compose it with a background scene with only the sky to achieve realistic lighting of the foreground subject. c) Comparison for depth-aware scene composition: SDEdit and Pair Diffusion generate unnatural compositions and distorts the identity in some cases. Our approach realistically blends the two scenes in a depth-aware manner, with consistent intra-scene illumination.

jects in the amodal bounding box as they have trained to primarily inpaint unoccluded objects with 2D bounding boxes. This is quantified with a poor CLIP-sim metric. Anydoor is able to generate consistent objects; however, it generates significant border artifacts (marked in red), resulting in unnatural composition. Our approach generates realistic compositions with accurate object placement (highest Clip-sim) superior identity preservation (highest Dino-sim) as compared to all the inpainting baselines. Further, the object is naturally placed at an accurate depth, as evident with higher $\Delta$ **depth** scores.

## 4.2 SCENE COMPOSITION

We compare Diffusion Compose for the task of depth-aware scene-composition with the following baselines: a) Image MPI + Harmonization (**Image MPI + H**), **Image MPI + SDEdit Meng et al. (2021)** for generating realistic compositions using MPI mask for foreground and background. Additionally, we also compare our method with **PAIR Diffusion Goel et al. (2024)**, which allows for localized control for a given masked region with a reference image. Specifically, we use the MPI mask to segment out the foreground and background regions and then pass the foreground and background reference images to PAIR Diffusion for generation.

Table 2: Comparison for depth-aware scene composition

| Method | LPIPS↓ | FID↓ |
|---|---|---|
| Image MPI + H | 0.036 | 132.6 |
| SDEdit | 0.395 | 106.24 |
| Pair-Diffusion | 0.45 | 140.54 |
| Ours | 0.263 | 123.32 |

**Metrics.** We measure the scene composition for visual quality, structure preservation of the foreground and background, and depth consistency. We report **FID** with the COCO dataset to quantify the realism of the scene compositions. To evaluate the structure and appearance preservation, we report the average **LPIPS** of the background and the foreground region. For realistic composition, LPIPS and the KID value should be low-achieving structure preservation with high realism.

**Analysis.** We present our results and comparisons in Fig. 6 and Tab. 2. Image MPI + Harmonization, achieves illuminates the foreground to improve blending, however still struggles with *cut-pasting* appearance (e.g. sofa scene) leading to improved structure preservation, but unrealistic generation (inferior FID score). SDEdit and PAIR-diffusion change the scene structure while generating consistent images in some examples.

Our method generates realistic depth-aware scene composition with accurate scene illumination while preserving the scene structure. Additionally, our method allows for realistic relighting of the scenes by providing different sky backgrounds. To analyze the depth consistency, we visualize the histogram of the input and the output scene in Fig. 7, which shows Diffusion Compose preserves the distribution of depth present in the foreground and background scene even during composition.

### 4.3 USER STUDY

Due to the unavailability of well-established metrics for the task, we perform an extensive user study to quantitatively evaluate our approach across multiple aspects. We perform a user study to evaluate Diffusion Compose for depth-aware scene editing. We evaluate object placement for the realism of the *placement*, *identity preservation*, and *depth consistency*. For the task of scene composition, we evaluate for the *realism* of the composition and *depth consistency*. The study was performed on 15 source images for each task and 40 volunteers participated with varied expertise in

image editing. We created 60 image pairs for object placement, and 40 pairs for scene composition, with each pair consisting of our generated output and a randomly sampled baseline. We divide this dataset into groups of 20 image pairs for separate analysis on each editing goal. Each user compared 20 pairs for each of the goals for the two tasks. The order of image pairs and the methods within each pair were randomized.

**Object placement.** The results of the study are presented in Fig. 8. Our method significantly outperforms all baselines in terms of realism, identity preservation, and depth consistency. Image MPI+Harmonization achieves better results for identity preservation, as it directly *cut paste* the object

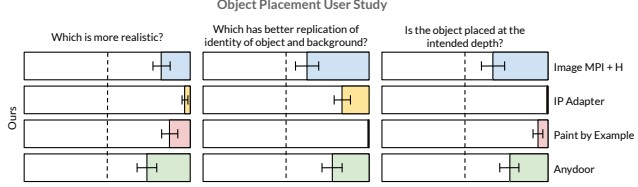

Figure 8: Object placement user study.

from the input image resulting in unrealistic generations. Paint by example and IP-adaptor performs poorly across all the three goals indicating the challenge of depth-aware placement task. Our approach excels in depth consistency metrics, indicating that our method effectively performs depth-aware editing while producing highly realistic images.

**Scene composition.** As indicated in user study (Fig. 9), Image MPI + Harmonization performs comparably to our approach for both goals. However, the harmonization model is specifically trained on a large scale dataset for the task of blending objects and foreground but the same baseline fails to perform well on the object placement task (Fig. 8). Our zero-shot approach achieves superior realism and depth consistency in the generation as compared to other scene composition baselines.

### 4.4 ABLATIONS

We ablate over the design choices with the task of scene composition in Fig. 10. We follow the same guidance parameters for the object placement task as well. Additional ablations are provided in the supplementary document.

**Guidance Timestesp.** We ablate over the timestep range from $0 - 50$ for applying the

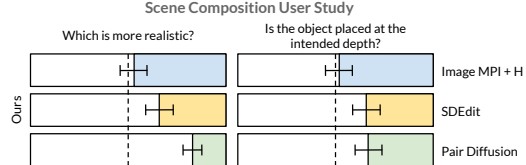

Figure 9: Scene composition user study.

multiplane feature guidance. Guiding only for small timesteps ($0-20$) results in significant structure changes for the foreground and background scenes. On the contrary, providing guidance for all the timesteps preserves the structure but leads to unnatural composition (lighting mismatch). We found

---

Figure 7: **Depth consistency in scene editing:** The initial depth regions in the composite image align with the foreground depth maps, while the later regions correspond to the background depth maps, indicating that the depth distribution is preserved in the composite image.

that guiding until an intermediate range of timesteps (0-38) and allowing the image to denoise freely for the remaining steps strikes a good balance, resulting in realistic compositions.

**Guidance Layers.** We ablate over the U-Net decoder features to guide the generation. Using all the decoder layers for guidance results in significant artifacts. We observe that guidance with the first decoder layers can significantly hurt the generation. Finally, we achieve a combination of layer 3 (weight $8.5$) and layer 2 (weight $0.2$) works well in most cases. Using only one of these layers resulted in subpar compositions.

**Guidance weight.** After finalizing the layers to be used for guidance, we tried different weights for the guidance factor. Specifically, we ablate over a guidance multiplier $\lambda$ for foreground guidance. Having a smaller $\lambda$ results in generating only a background region, we achieve a good composition with $\lambda = 1$. Notably, $\lambda$ is also a control parameter that a user use to control the effect of the background on the foreground scene.

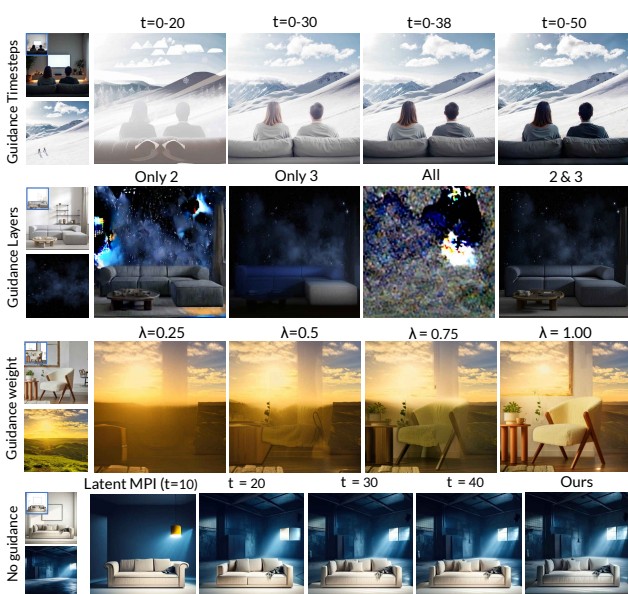

Figure 10: Ablation for scene composition guidance

**No guidance.** We ablate against directly compositing multiplane representation of diffusion latent at timestep $t$ and allow the model to freely denoise from $t$ to $0$ timesteps. With lower timesteps, the generation is coherent but the identity of the sofa(Fig. 10) is significantly changed, however, when blended at a later timestep the illumination of the foreground is not adapted. The guidance based approach enables us to slowly align the latents at each step of denoising to achieve identity-preserving and coherent compositions.

## 5 CONCLUSION AND DISCUSSION

**Conclusion.** In this work, we propose Diffusion Compose an efficient zero-shot framework for depth-aware scene editing with Text-to-Image diffusion models. We leverage multiple scene representations and incorporate them in the generation process of diffusion models. Precisely, we propose a novel multiplane feature guidance approach to guide the generating diffusion latents towards the target edit while preserving the scene structure. We demonstrate the effectiveness of our depth-aware editing framework with the task of realistic scene composition and 3D-aware object insertion. Diffusion Compose generates highly realistic scene editing results without a need for retraining. We believe our work provides a new perspective on augmenting the capabilities of Text-to-Image diffusion models for 3D-aware editing.

**Limitations.** Our framework is based on pretrained Text-to-Image diffusion models and inherits the limitations and biases of the base model, such as geometrically inconsistent shadows and perspectives in some cases. Further, as we are applying guidance at each step of denoising, the proposed method is slower than generation from the base Text-to-Image diffusion model.

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

## A  APPENDIX

In this appendix, we provide more details about the dataset used for the evaluation of object placement and scene composition and implementation details and hyperparameters used for both the task. We also conduct a user study, and its details will be explained in detail, along with a few additional results and comparisons for both scene composition and object placement.

## B  DATASET DETAILS

**Scene composition.** We collect background images from the SSharmonization dataset Jiang et al. (2021), and for foreground images, we take a variety of images with different lighting from Google images. Our dataset consist of around 2844 images with 80 background images and 36 foreground images. To get the foreground mpi mask, we manually do annotation to find the best MPI plane where we can get a meaningful foreground region that can be composed with other background images.

**Object Placement.** Our object placement evaluation dataset consists of 491 background, object paired images. All of these are collected from Google images consisting of outdoor and indoor scenes with various kinds of occlusions. We manually annotate each pair to get the object bounding box and the MPI depth layer where the object can be placed with proper occlusion.

## C  ABLATION

Since this feature guidance has a lot of hyperparamters, we will explain their role and the choice we made for choosing specific values.

**Scene composition.** With respect to which layers to use for guidance and what timestep to give guidance, we have shown the result in Fig. 10, which clearly explains our parameter choice. But in regard to the MPI-based rendering in latent space, there are other choices like direct latent MPI rendering or latent MPI guidance. In this section, we show why feature guidance is used and how it is better at scene composition compared to the other mentioned alternatives. In direct latent MPI guidance, we could just cup-paste the foreground and background DDIM latent at some timestep and let the model denoise it as usual. But from b) in Fig. 11 we can see that doing this MPI rendering in latent space at an earlier timestep has some lighting change but the scene identity is completely lost and doing this at a later timestep preserves scene identity but there is no composition, it is similar to image

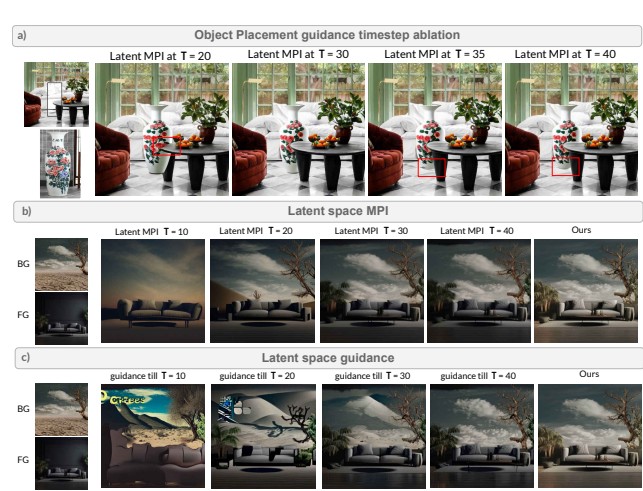

Figure 11: a) Ablation for timestep is used for guidance in object placement task; the other two rows have ablation for alternative latent space MPI-based composition methods. b) Ablation for Latent space MPI for scene composition, c) Ablation for Latent space guidance method for scene composition

MPI with no lighting change. Another option is to give latent space guidance for the foreground region and background region, similar to how we give it in feature space in our method. We can give this guidance from the beginning till a particular timestep. In Fig. 11 c) part, we have shown the result for latent guidance till different timesteps, and as we can see in the earlier timestep, we have major identity loss, and at the later timestep, it is similar to the image space MPI, but with a little lighting change. Compared to our feature guidance both these alternative latent MPI methods fail to do scene composition.

**Object Placement** In the depth-aware object placement task, since inpainting distorts the foreground region, we use our guidance to preserve the foreground region based on the MPI masks. In this, our guidance tries to preserve the foreground, and Anydoor's additional conditioning tries to erase the foreground and paint the whole object in the foreground. As we can see, both of these go against each other, and since the guidance is weak compared to edge conditioning in Anydoor, initially, we do latent space MPI rendering as mentioned in Sec. 3.4 and then give guidance to foreground region. And there is a choice to do this latent MPI rendering at which timestep. We saw that doing this at an earlier timestep gives more freedom to Anydoor, and thus, it doesn't put the object with proper occlusion as we can see in Fig. 11 and doing this at a later timestep can cause artifact at the mpi mask foreground edge.

## D  IMPLEMENTATION DETAILS

Our guidance method is based on Pandey et al. (2024b). As we can see from 10, using any layer other than the last feature layer causes major artifacts in generated images. Using only the last layer features for guidance, doesn't cause any artifacts, but image appearance is lost for background and foreground. So, we give high weightage to the last layer feature and very low weightage to the last before layer for better identity preservation. For scene composition, we start from the inverted latent of the background image and give guidance to the foreground and background layers according to their corresponding features. Starting from background layer latents causes the scene lighting to be inherited from the background scene. We give guidance from 0th timstep to 38 similar to diffusion handles. We also show in 10 that giving guidance till 50th timestep causes it to look similar to cut paste without any lighting change.

The same layered guidance is used for object placement, and we use the same parameter used for scene composition except the number of optimization per step. For object placement, we only optimize for 3 steps per iteration since the inpainting model already does well in preserving the appearance of the features outside of the bounding box. At T th timestep we perform a cut paste of latents according to the MPI layer mask, then for the rest of the generation, we give guidance to preserve the foreground. If the T value is early, then Anydoor edgemap condition becomes strong and puts the object in the foreground, and if T is late, then image looks similar to latent cut paste with artifacts at the border. We qualitatively found T=30 timstep to be working best for most cases.

The time taken to generate a single image for object placement is 60 seconds, and for scene composition, it takes around 86 seconds.

## E  USER STUDY

We conduct a qualitative comparison of our Object Placement method against four baselines: Image MPI + H, IP Adapter, Paint by Example, and Anydoor. The evaluation focuses on three key goals: scene realism, identity replication of the placed object and background, and accurate placement at the intended depth. To assess these goals, we carried out a user study on 15 edits across 15 images from our Object Placement dataset. Each goal was evaluated separately by presenting users with pairs of images and asking them to select the one that better met the specific goal. A total of 60 randomized image pairs were generated, with each pair comparing a result from our method to a corresponding result from a randomly chosen baseline. These pairs were divided into three groups of 20 pairs each, corresponding to the three goals. The study involved 40 participants with varied experience in image editing, who evaluated 20 pairs for each goal, resulting in 800 data points per goal and 2400 data points in total. To mitigate bias, the order of image pairs and the methods within each pair were randomized.

For Scene Composition, we compare our approach against Image MPI + H, SDEdit, and Pair Diffusion, focusing on two goals: realism and depth consistency. Using a subset of 15 edits across 15 images from our Scene Composition dataset, we generated 40 image pairs, split evenly across the two goals. The same 40 users participated in this evaluation, generating 800 data points per goal, for a total of 1600 data points.

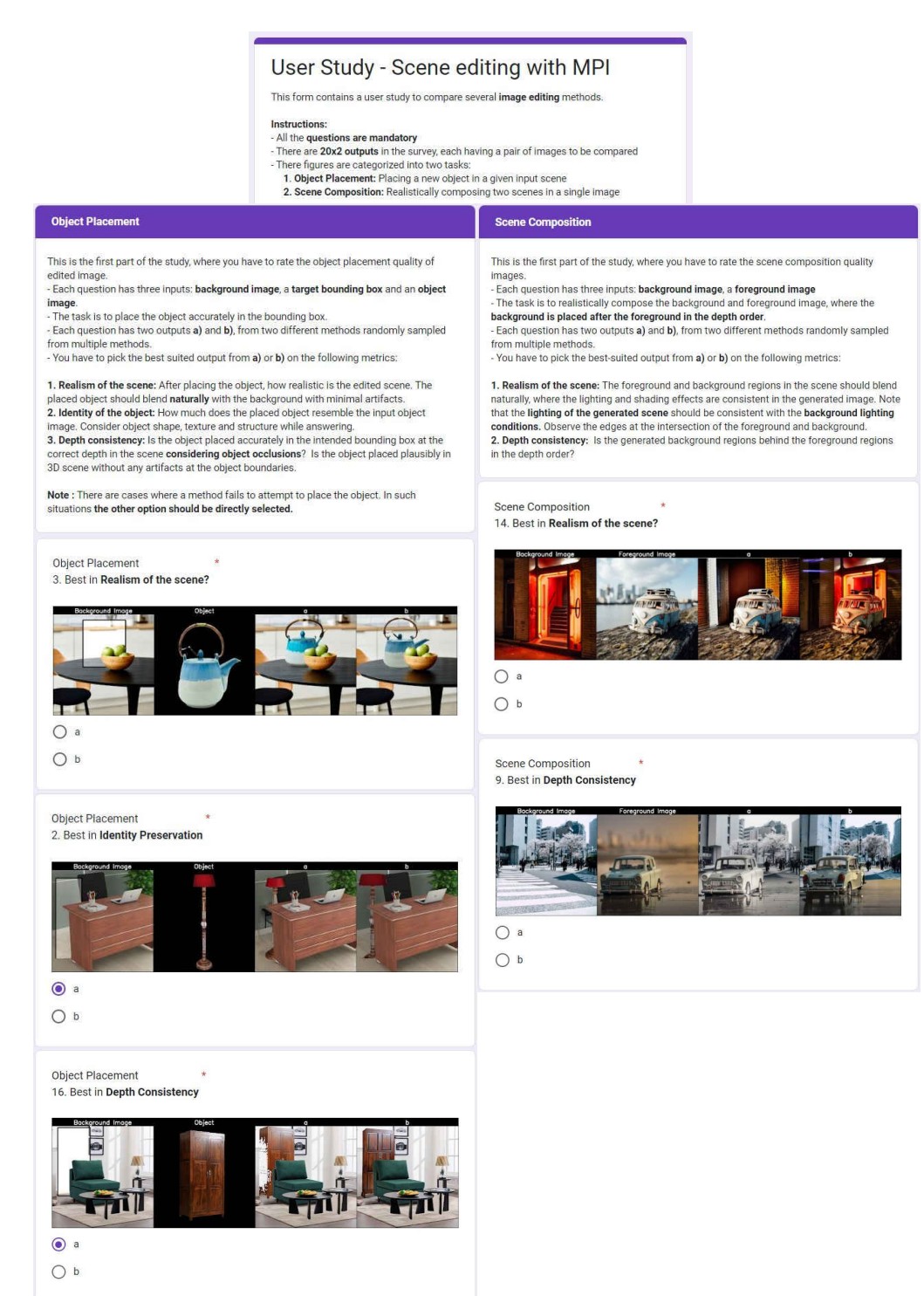

Figure 12: **User Study Screenshot.** We asked three types of questions on Object Placement(left): Realism of the generated image, Identity of object and background, Depth accuracy and two types of questions on Scene Composition(right): Realism and Depth Accuracy.

| Foreground | Object | Image MPI +Harmonization | IP-Adapter | Paint by example | Anydoor | Ours |

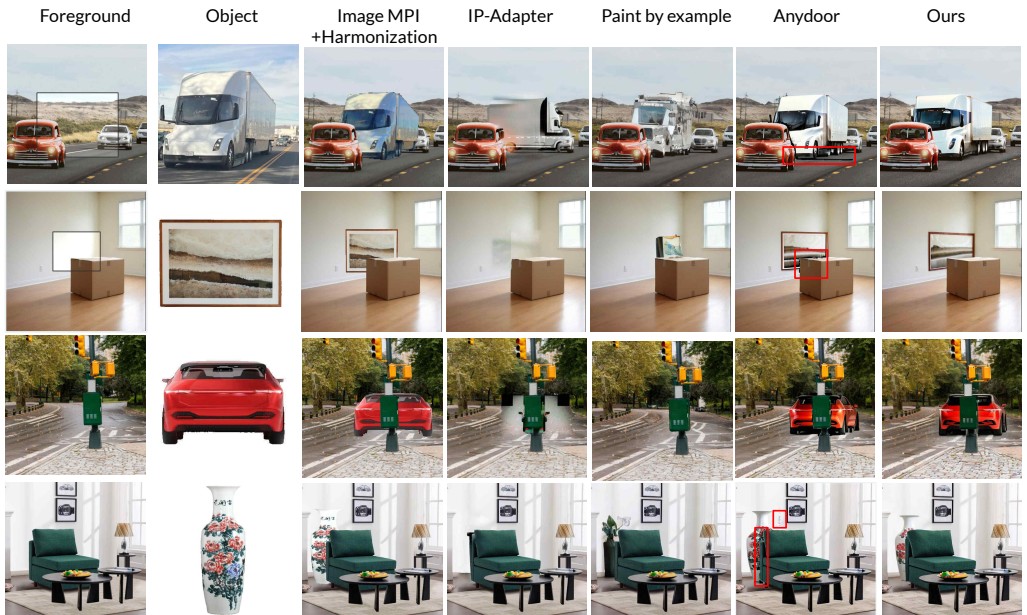

Figure 13: Comparison of depth-aware object placement: Image MPI + Harmonization results in an unnatural 'cut-paste' appearance for the inserted object. Inpainting models IP-adapter and Paint by example struggle to insert objects with consistent identity given the amodal bounding box. Anydoor achieves decent placement but has significant artifacts at the mask border (marked in red). Our method achieves realistic object placement while preserving the object identity and scene consistency.

# F ADDITIONAL RESULTS

In this section, we provide additional results for comparison with other baseline methods on both tasks and additional results for various scenes by our method.

Figure 14: a) Results for depth-aware object placement. b) Our method can also place the given object at multiple locations in a depth-consistent manner

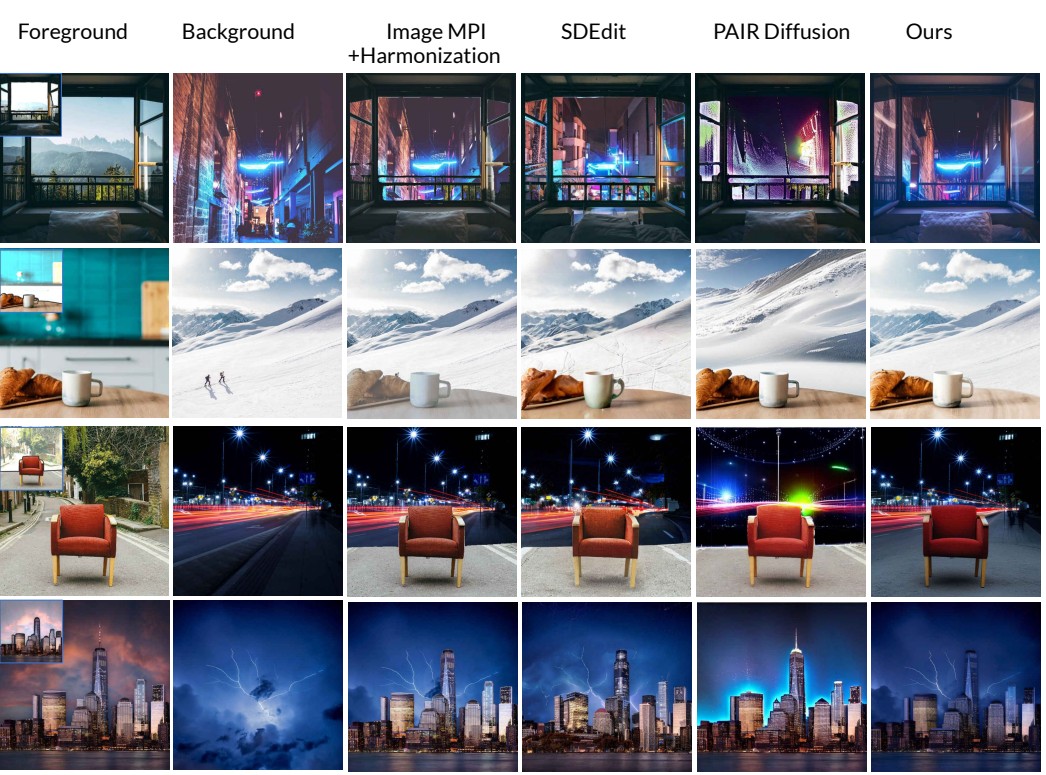

Figure 15: Comparison for depth-aware scene composition: SDE edit and Pair Diffusion generate unnatural compositions and distort the identity in some cases. Our approach realistically blends the two scenes in a depth-aware manner with consistent intra-scene illumination.

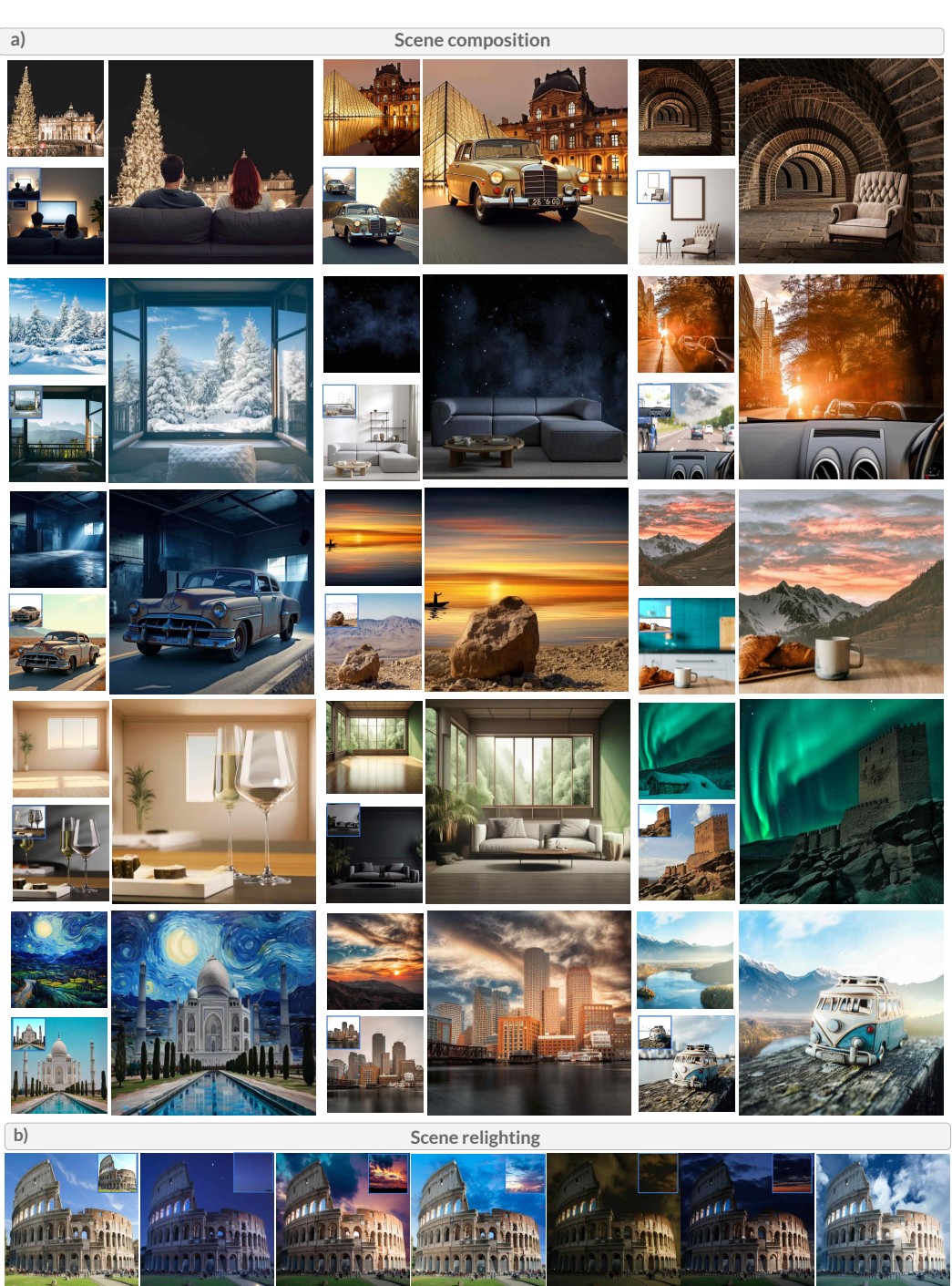

Figure 16: a) Results for scene composition b) Given a foreground scene, we can compose it with a background scene with only the sky to achieve realistic lighting of the foreground subject.

