# OpenReview forum: "Diffusion Compose: Compositional Depth Aware Scene Editing in Diffusion Models"
_ICLR.cc/2025/Conference — ICLR 2025 Conference Withdrawn Submission_

### Official Review · Reviewer_HAi2 · 2024-10-28

**Soundness:** 3
**Presentation:** 2
**Contribution:** 3
**Rating:** 5
**Confidence:** 4

**Summary:**

The authors propose a method, Multiplane Feature Guidance (MPG), that allows depth aware scene editing via diffusion models. They propose a framework, Diffusion Compose, that integrates MPG with specific implementation details which allows scene composition and object placement in a depth aware manner. The authors include extensive qualitative/quantitative results that demonstrate the effectiveness of their proposed method.

**Strengths:**

1. Authors incorporate Multiplane imaging, a classic CV method into the generative framework in an elegant way. The intuition behind MPG is simple yet effective, and the 'intuition' snippets provided by the authors in section 3.3 shows the consideration of the authors for readers.
2. Compared to baseline methods/models, qualitative results of the proposed method are impressive. Results of scene composition are especially high fidelity, in the sense that foreground objects are relighted/blended with the given lighting of the background image.
3. Authors are the first to perform depth aware object placement with a single exemplar object/background image. Because this task has not been done before, authors define new baselines that are reasonable to compare their proposed method.

**Weaknesses:**

1. Details about the proposed method can seem misleading. Authors should mention that the images used for editing (foreground/background objects) are generated images, and the intermediate latents used to generate said images should be saved/stored in order to perform editing.

2. Authors claim that their proposed method is applicable for text to image diffusion models, but this claim is also misleading. For implementations, the authors use Stable Diffusion v2-depth and Anydoor, which are inherently trained to accept an additional depth map as the input condition. The proposed methods for object insertion and scene composition seems to rely on the model's ability to accept additional depth maps. Authors should either rephrase their claims or include experiments that use external plug and play modules that allow depth map inputs.

3. Additionally, details on how the prompts are curated for the experiments are missing. Given a background image A, object image B, and the text prompts used to generate said images, what would be the edit text prompt?

4. Detail on how the multiplane representations are curated seem lacking. What monocular depth estimation model was used for the experiments? It would be better if the authors could show how the multiplane representations are actually extracted using the depth map of an image next to figure 2, it would clarify their method.

5. The detailed method seems to hinge on the depth map representations of the foreground/background image. However, depth maps extracted from monocular depth map estimators are inherently relative. For example, the depth map extracted from an image of a very far away object do not reflect the absolute distance of the object from the camera. How would this translate to scene compositions of objects very far away to the camera with background objects very close?

6. Although the inherent method behind object insertion/scene composition is the same, the qualitative results seem to vary. Identity preservation for objects seem very good for scene composition, but for object insertion, the object identity is not well preserved. The authors should address this, providing insight on why this is the case.

7. Although the clarity of the paper is sound, there is room for improvement. Namely,
	1. The authors should settle for a single term for their task. Object insertion/placement seems to be used interexchangeably.
	2. In section 3.3, authors define color as $c_i$ and opacity as $\alpha_i$, but for the rest of the paper, $c^i$ and $\alpha^i$ is used.
	3. Including bold/italics on the best/second best performing metrics in tables improve readability.

Overall, the paper itself proposes a novel method for depth aware editing, but its novelty is shadowed by the lack of crucial details in its implementations. I encourage the authors to resolve the concerns above.

**Questions:**

See above.

---

### Official Review · Reviewer_Kwdj · 2024-10-29

**Soundness:** 2
**Presentation:** 2
**Contribution:** 3
**Rating:** 5
**Confidence:** 4

**Summary:**

The paper proposes Diffusion Compose, a zero-shot approach for depth-aware scene editing with text-to-image diffusion models. Unlike traditional 3D-aware editing that primarily focuses on single-object control, Diffusion Compose enables scene composition and object insertion by using a multiplane representation that divides scenes into fixed-depth planes. This technique allows for individual layer adjustments and natural reassembly with depth consistency and occlusion. Diffusion Compose applies multiplane feature guidance, incrementally aligning layer features at each denoising step, which preserves structural and lighting consistency. Validated on tasks like object insertion at specific depths and scene composition, Diffusion Compose outperforms existing methods in realistic depth-aware editing, offering valuable applications in areas like advertising and game design, where natural layering is crucial.

**Strengths:**

- The proposed model employs multi-plane imaging to separate representations, subsequently utilizing these distinct representations within a diffusion guidance.

- Through the use of guidance, the model achieves depth-aware image editing in a zero-shot setting.

- Extensive experiments presented in the paper demonstrate that the proposed model outperforms existing methods in depth-aware object placement.

**Weaknesses:**

- While the qualitative results of the ablation studies highlight the benefits of feature guidance, additional quantitative analysis would provide a clearer demonstration of the advantages offered by multi-plane feature guidance. Please report DINO-sim, KID, delta_depth, Clip-sim for object placement as shown in Table 1 and report FID and LPIPS for scene composition as shown in Table 2.

- The method heavily depends on the additionally provided depth information, as Anydoor generates the mask independently using the Sobel filter. This suggests that additional information may be necessary to further enhance performance.

- The use of multi-plane feature guidance is notably effective in enhancing the geographical consistency between foreground and background elements. However, based on the results of Latent MPI, it appears that lighting harmonization is primarily attributed to the capabilities of the diffusion model itself, rather than to the proposed method.

- The diffusion timesteps and optimization layers in the proposed method are selected empirically, which may impact its generalizability and applicability for both representation learning and image editing tasks. A more rigorous analysis of these hyperparameter choices, even with a brief theoretical foundation, could provide insights into how they affect the model's performance. If possible, please suggest any theoretical motivations or analysis for the chosen value.

- Figure 2 should be placed above Figure 3, and the caption of Figure 3 should explain Latent MPI, with labels (a) and (b) marked in the figure. It is unconventional for Figure 2 to be placed below Figure 3, which disrupts the logical flow of the presentation. Additionally, the caption for Figure 3 lacks sufficient detail to adequately explain the figure, and some labels are mismatched. Please ensure all figures are properly labeled and referenced in the text to enhance overall readability.

- The absence of parentheses for citations throughout the paper hinders readability, making it challenging to distinguish references from the main text. Please write parentheses for citations and use \citet{} and \citep{} following “Formatting Instructions for ICLR 2025 Conference Submissions”.

**Questions:**

- The performance gains appear to stem from the well-structured masks that incorporate multiple depths. Could this type of mask be applied to Anydoor to enhance its performance? In the paper, it said that it uses the depth-aware Anydoor.

- Would it be possible to visualzie the 3d-reconstructed scenes after object replacement and scene composition? Since this model only outputs the image, is it truly accounting for depth?

- How would depth-aware editing perform on scenes with significant depth variations? Could combining a foreground object from an outdoor scene with a background from an indoor scene be feasible?

---

### Official Review · Reviewer_DENf · 2024-10-29

**Soundness:** 3
**Presentation:** 3
**Contribution:** 2
**Rating:** 5
**Confidence:** 3

**Summary:**

This paper studies a novel problem of performing depth-aware image editing, achieving considerable results.

**Strengths:**

(1) The studied problem is novel and can be applied in many scenarios.

(2) The editing is performed in a zero-shot manner with slight training costs.

(2) The paper is overall well-written.

**Weaknesses:**

(1) In Table.1, the performances are worse than Image MPI+H except for the CLIP score. Besides, table.2 shows that the proposed method does not present a significant advantage over Image MPI+H.  In addition, Although the author demonstrates that  MPI+H suffers from copy-paste, the visualization in Figure 5 is insufficient.

(2) It is unfriendly for users to provide a specified depth during editing.

More see the Questions part.

**Questions:**

(1) Equation 5 employs a loss to close the feature gap between the edited image and reference images. Why need to introduce loss-based optimization, in my view, it can be accomplished with a simple linear combination of two features. In addition, how about the effect of different number of optimization iterations  (like increasing 5 to 8) in each timestep?

(2) It seems the method heavily relies on the quality of depth plane splitting, it is unclear how to obtain the x^depth in Line 192. If it is obtained from an off-the-shelf depth estimation model? How about the effect of depth estimation on final editing quality?

(3) If the opacity in alpha blending(Equation 3) is important? Why not to directly calculate on typical 2D masks? Just like an image segmentation mask without considering alpha.

(4) Line 266, what is the format of given relative depth, if the user can provide a sentence to describe the relative position for object insertion?

---

### Official Review · Reviewer_axjs · 2024-11-03

**Soundness:** 3
**Presentation:** 2
**Contribution:** 2
**Rating:** 5
**Confidence:** 4

**Summary:**

This paper proposes a method for realistic depth-aware scene editing using text-to-image diffusion models. The technique enables accurate control over object placement at specific depths and seamless blending of multiple scenes. By using a multiplane feature guidance approach, it preserves structure and lighting in 3D edits without additional training. Experimental results show that this work produces high-quality, depth-consistent edits, making it effective for applications requiring realistic 3D scene manipulation, such as visual effects and game design.

**Strengths:**

1. This paper tackles the interesting issue of precise depth-aware control in text-to-image (T2I) diffusion models, essential for applications like game design and visual effects where realistic layering is critical.
2. Using multiplane feature guidance, the authors introduce a smart way to handle depth-based object insertion and scene composition. This approach aligns the latent features of the source with the target at each denoising step, preserving both realism and structure. The technique is effective and requires no additional training, making it versatile and practical.
3. The results showcase realistic, depth-consistent edits, outperforming baselines. The authors provide a clear analysis of each module’s role through visual and quantitative comparisons, demonstrating how each component contributes to the overall quality.

**Weaknesses:**

1. Insufficient Discussion on Related Work: The paper does not sufficiently discuss relevant prior work in layered object and scene compositing, a concept that has been well explored in recent literature. For instance, LayerDiffusion (Li et al., 2023) proposes a method that optimizes different compositional layers separately, which closely parallels the layer-wise approach used in this study. Similarly, LayerDiff (Huang et al., 2024) decomposes the generation task across distinct layers to handle compositional generation more effectively. The authors should provide a more comprehensive discussion of these methods and conduct a comparative analysis to clearly delineate the advantages and limitations of their proposed approach relative to these existing techniques.

2. Limitations with Object Interactions in Layered Compositions: The results primarily showcase examples of simple object insertions, where each object is placed within a single, isolated layer. However, real-world scenarios often involve more complex interactions between objects across layers, such as the case of a person hugging a dog or overlapping occlusions. It remains unclear how the proposed pipeline would handle such interactive or occlusive scenarios effectively. Additional examples or an analysis addressing these cases would greatly enhance the robustness and applicability of the proposed method.

3. Inconsistent Object Insertions: The paper does not address inconsistencies observed in object insertions. For instance, in Figure 1, the inserted lamp changes color across different frames or compositions, indicating a lack of consistency in maintaining visual coherence. This inconsistency could undermine the practical reliability of the method, especially in applications where uniformity in appearance is crucial. A discussion on potential causes and solutions for maintaining visual consistency in object attributes (such as color) across compositions would add value to the paper.

[1] Li P, Huang Q, Ding Y, et al. Layerdiffusion: Layered controlled image editing with diffusion models[M]//SIGGRAPH Asia 2023 Technical Communications. 2023: 1-4.
[2] Huang R, Cai K, Han J, et al. LayerDiff: Exploring Text-guided Multi-layered Composable Image Synthesis via Layer-Collaborative Diffusion Model[J]. arXiv preprint arXiv:2403.11929, 2024.

**Questions:**

1. Should the parameter \alphaα be employed in Equation (5) to filter out unnecessary optimization steps?
2. In Figure 4, there is an extraneous “the” before “t” in the second line of the caption.

---

### Note · Authors · 2024-11-14

**Comment:**

I wish to withdraw my submission from ICLR

**Withdrawal Confirmation:**

I have read and agree with the venue's withdrawal policy on behalf of myself and my co-authors.